# Environmental Activism and Big Data: Building Green Social Capital in China

**Jing Xu** [1,*]  **and Huijun Zhang** [2,*]

1   Center for Energy and Environmental Policy, University of Delaware, Newark, DE 19716, USA
2   Department of Physics, The Hong Kong university of Science and Technology, Clear Water Bay,
    Hong Kong 87P8+H5, China
*   Correspondence: xujing@udel.edu (J.X.); hjzhang@ust.hk (H.Z.)

**Abstract:** The rapid development of information and communication technologies, coupled with the significant progress in the areas of environmental policy and public participation, has led to the advent of environmental big data in China recently. This article applies social capital theory as an analytical lens to shed light on how Chinese environmental non-government organizations (ENGOs) adopt big data to promote environmental governance. This study conducts case studies focusing on two ENGOs: The Institute of Public and Environmental Affairs (IPE) and Green Hunan. Combining a qualitative approach with quantitative analysis, this research examines two big data-induced initiatives: The first involves green supply chain management in the IPE, brand-sensitive multinational corporations (MNCs), and Chinese suppliers of the MNCs, while the second involves the mobile data-based Riverwatcher Action Network of Green Hunan and numerous volunteers nationwide. This study found that big data adoption by ENGOs contributes effectively to building green social capital, including social networks and pro-environmental social norms. Green social capital has important implications for governance in terms of fostering coordination and cooperation across the boundaries of the public, private, and voluntary sectors. This study highlighted the finding that empowerment by big data helps Chinese ENGOs play the role of a change agent in sustainability transitions.

**Keywords:** big data; environmental NGOs; social capital; China

## 1. Introduction

Information and communication technologies (ICTs) have had a substantial influence on Chinese environmental non-government organizations (ENGOs). The development of environmental activism and the adoption of ICTs are by no means mutually exclusive; in fact, these aspects are embedded or mutually constitutive [1,2]. The invention of the internet was a major and ground-breaking step in the recent history of ICTs [3]. The year 1994, when China was first connected to the World Wide Web, also marked the emergence of China's first ENGO: Friends of Nature [4]. Since then, with the diffusion of the internet, web-based ENGOs quickly sprouted in the country [5]. Greener Beijing, one of China's oldest web-based ENGOs, launched the Save the Tibetan Antelope campaign in 1999. It created the website, Save the Tibetan Antelope, and built online alliances with numerous organizations across China [4]. This internet-based campaign is one of China's first large-scale environmental collective actions involving internet participation. ENGOs often utilize three of the many various functions of the internet, namely Web sites, mailing lists, and bulletin boards [5,6]. These internet technologies: Enable ENGOs to survive and operate with minimal financial resources; help them overcome certain political constraints; assist them in building virtual network connections at home and abroad; empower them to produce discourse and circulate environmental information; facilitate the creation of an organizational

presence, and improve visibility, all of which have led to the formation of green public sphere in China [7,8].

ICTs later advanced to another evolutionary stage, commonly described as Web 2.0 or social media, in the late 1990s and early 2000s [3,9]. This includes blogging, social networking, Wikis, and a host of other applications [10]. In general, it refers to a collection of technologies that are: Interactive and bi-directional; use the internet as a platform for creating network effects; emphasize user-created content; allow users to efficiently generate, disseminate, share, and edit/refine informational content [9,11,12]. These technologies are attractive to NGOs because the interactivity, decentralized structure, and interaction ties associated with these technologies have the potential to improve NGOs' capacity for civic engagement [3,13]. Chinese ENGOs were late in adopting this type of ICTs. They began to embrace and employ Web 2.0 technologies to engage stakeholders and influence policy in the early 2010s. The microblogging-based PM$_{2.5}$ campaign launched by ENGOs in the fall of 2011 offers a vivid example. Fedorenko and Sun observed that microblogging platforms play a decisive role in mobilizing millions of citizens, particularly with regard to engaging public figures and governmental agencies in the online activism against air pollution [1]. The campaign prompted the Chinese government to enact changes in air pollution monitoring policy, mainly as a result of the use of the interactive technologies and the ability of ENGOs to adapt and innovate their communication and mobilization strategies.

Technology is ever-changing. While Web 2.0 will continue to evolve, ICTs have moved forward again to Web 3.0, the Semantic Web [3,14]. According to World Wide Web Consortium, the Semantic Web, which is a web of data, provides a common framework that allows data to be shared and reused across application, enterprise, and community boundaries. The advent of Web 3.0 is intimately connected to the rise of big data [15]. Web 3.0 fits well with the growing role of data and data-informed activities in NGOs [3]. Big data are characterized by three Vs: high-volume, high-velocity, high-variety [16]. With the rapid development of smart sensor technologies, wireless transmissions, network commutation technologies, mobile devices, cloud computing, and the Internet of Things, environmental data are being produced and collected at an unprecedented scale in China. The massive real-time data on air quality and water quality provide a small glimpse into the volume and velocity of big data in the country's environmental sector [17]. At the national level alone, 1436 air quality monitoring stations exist in 338 cities across China to capture and disclose real-time data per hour for key air pollutants, such as particulate matter 2.5 (PM$_{2.5}$), PM10, and sulfur dioxide (SO$_2$). Moreover, the public also has access to the real-time four-hourly data of surface water quality from 1940 monitoring stations spanning across China's 978 major rivers and 112 major lakes [18,19]. If the monitoring stations at the local level are included in this count, the volume of environmental data grows exponentially. The advent of big data has opened up new opportunities and ideas for Chinese ENGOs to improve environmental information availability and expand public awareness and civic engagement in fields such as pollution control. Further, big data applications have also become attractive to the ENGOs of other countries and have helped them create innovative strategies to address environmental challenges. For example, ENGOs in the UK have adopted big data in various areas, such as biodiversity monitoring, and they have identified numerous opportunities to leverage big data use towards sustainability [20]. An international ENGO based in the US has used big data to monitor changes in the world's forests and to tackle deforestation [21]. In Vietnam, ENGOs are adopting big data-based approaches to improve air quality monitoring and governance [22].

ENGOs can employ big data technologies, such as databases, data mining tools, and data analytical methods to "extract value from very large volumes of a wide variety of data by enabling high-velocity capture, discovery, and/or analysis" [23] (p. 1), and to provide useful knowledge that informs more intelligent decisions and effective actions [24]. As Williams noted, "data can be used for civic action and policy change by communicating with the data clearly and responsibly to expose hidden patterns and ideologies to audiences inside and outside the policy arena" [25] (p. 181). However, while environmental big data have emerged and are on the rise, particularly in China, the existing literature

rarely discusses how ENGOs collect, analyze, and derive knowledge and insight from big data. Moreover, little attention has been paid to how ENGOs conduct data-driven initiatives to harness big data's potential and develop new capacities in the pursuit of their missions. To fill this gap, this study seeks to address two questions in the Chinese context: How do ENGOs pursue big data-based activities? What are the implications of big data adoption by ENGOs for China's environmental governance?

## 2. Theoretical Foundation

Since the 1990s, the concept of social capital has gained prominence in social science research [26]. It has been extensively applied to explore the use and social impacts of ICTs in various areas such as human development [27–30], business management [31], social inclusion [32], anticorruption [33], and community development [34]. These studies have consistently demonstrated the usefulness of the concept to elucidate the role of ICTs and the effects of their adoption on society and social change. Therefore, social capital also provides a useful analytical and theoretical tool for society to understand the use of big data by ENGOs and its governance implications.

Social capital "is rooted in social networks and social relation" [35] (p. 31) at the organizational and individual levels. Putnam defined social capital as "features of social organization such as networks, norms, and social trust that facilitate coordination and cooperation for mutual benefit" [36] (p. 67). Social capital can be disaggregated into three categories: bonding, bridging, and linking [37]. Bonding social capital refers to cooperative relations and connections among homogenous groups of people [38]. Bridging social capital comprises relations involving respect and mutuality between socially heterogeneous individuals or groups who are dissimilar in some socio-demographic sense but more or less equal in terms of their status and power [38,39]. Therefore, bridging social capital, in essence, can be considered as a horizontal pattern of linkage. Linking social capital, by contrast, describes relations among individuals or groups who interact across explicit vertical power differentials [37,40]. Despite various types of social capital, they all contain some aspect of social structure [41,42]. They have two characteristics in common in terms of effects: They all constitute valuable resources and assets for social actions and they enhance the abilities of all participants to act together more actively and effectively to resolve collective-action problems [43–45].

The central idea of social capital is that individuals or groups engage in interactions and networking to pursue shared objectives [35]. Social capital is a multidimensional concept that includes three distinct yet highly interrelated dimensions: structural, relational, and cognitive capital [45]. Relational social capital often refers to the actual and potential resources acquired through personal connections people have developed with each other through a history of interactions, which are generated and enjoyed at the individual level, such as trust, intimacy, liking, and respect among individual actors [44–47]. This research concerns green social capital derived from ENGOs at the organizational level, where structural and cognitive social capital play major roles. Therefore, this research focuses on these two dimensions only—the structural and cognitive dimensions—as they are directly relevant to how big data adoption in ENGOs leads to the formation and accumulation of green social capital in China. Social capital is "embedded within, and derived from the network of relationships possessed by an individual or social unit" [45] (p. 243). Essentially, social capital is centered around a network-based structure within which actors are located [46,47]. Networks are comprised of a set of social interaction ties among different actors that are organized and governed to initiate and facilitate information sharing, resource exchange, communications, and interactions [29]. Structural social capital represents the overall pattern of such ties [35,45]. It consists of two core aspects: the structure of the relationships and the content of the relationships [46]. The structure of the relationships covers a wide range of characteristics regarding the configuration of linkages between people or units, including: The presence, number, and strength of social interaction ties; the number and network locations of actors; the density, intensity, and hierarchy of social interactions [48–50]. Social interaction ties provide the necessary conditions for the transmission and use of information, and create opportunities to encourage and enable actors to interact with one another [51]. Therefore, social interactions and collective actions

through the network of social relations constitute the content of the relationships, and are preconditions for the development and maintenance of social capital [45,52].

The cognitive dimension of social capital is primarily concerned with the shared understanding and interpretations among actors who play a motivational role in fostering and facilitating collective actions of people or groups within a network [45,53,54]. The development of social capital requires a degree of shared meaning and understanding among actors about their common goals and the ways that enable them to effectively draw on their collective knowledge and capacity to achieve mutually compatible ends [37]. Two facets exist with regard to cognitive capital: shared goals and shared culture. Inkpen and Tsang argued that "shared goals represent the degree to which network members share a common understanding and approach to the achievement of network task and outcomes" [55] (p. 153). The establishment of shared goals can guide the nature, direction, and magnitude of the actions of network members [56]. Further, shared culture refers to a common set of rules and norms, both implicit and explicit, which govern appropriate behaviors and perceptions of members in the network [55]. The generally accepted rules and norms help create a shared context for network members and provide them with proper ways of acting in a social system, contributing to a harmony of interests and increasing the overall level of commitment to collective objectives. Overall, where congruent goals and behavioral norms both exist within a relationship, network members can more easily and effectively coordinate their actions and adapt their behaviors to the demands of a common task [57].

## 3. Method: Design of Multiple-Case Studies

Grounded on the theoretical lens of social capital, the purpose of this research is to provide an in-depth examination of how structural and cognitive social capital are built as a result of big data adoption by ENGOs. The "how" and "why" questions to investigate contemporary phenomena within the real-life context are more explanatory and tend to favor a case study as the preferred research method [58–60]. Two case studies are employed for this study. Social capital resides in social actors' relations. This actor-centered notion provides guidance and direction for defining and selecting cases. Chinese ENGOs often adopt big data-enabled activities to deal with two key actors: the general public and the business communities. Social capital can arise from both ENGO-business relations and ENGO-public networks. As a result, the following two ENGOs are selected to specifically illustrate how social capital is created in these two settings: The Institute of Public and Environmental Affairs (IPE) and Green Hunan.

IPE, a Beijing-based ENGO, was founded in 2006. From the beginning, IPE has been dedicated to developing data-based approaches to make multinational corporations (MNCs) and their polluting suppliers participate in environmental issues. Its signature initiatives—an environmental map database and a green supply chain network—are good examples to describe and explain how social capital is formed among ENGOs and companies. Green Hunan was established in the Hunan province in 2007. It has focused on building the Riverwatcher Action Network to address water pollution problems. The core strategy of the network is to engage volunteers to produce and publicize environmental monitoring data on a large scale and in a timely fashion. With the help of technology, the network has expanded from focusing on only one river in 2011 to covering all of the four major rivers of the Hunan province in 2015, and later, it grew into a national model in 2019. Therefore, Green Hunan is an appropriate case study to analyze the development of social capital among ENGOs and the public.

The data were collected from archival documents, the websites of IPE and Green Hunan, and other grey literature produced by the two ENGOs, including newsletters, annual reports, technical reports, project and activity reports, and academic publications. This data collection technique was complemented by semi-structured interviews conducted in 2018–2019 with two representatives of IPE and Green Hunan. ENGO representatives were interviewed and asked questions focusing on social relations and interactions in the green supply chain network and the Riverwatcher Action Network, as well as network members' attitudes and motivations. As a result, the data set in this study

comprised two types of data: quantitative and qualitative data derived from transcripts of interviews and a wide range of documents and records. Based on social capital theory, this study employed a mixed quantitative and qualitative approach for the data analysis. It consists of two parts in each case study: structural social capital and cognitive social capital. The data sources and methodologies for the data analysis are shown in Table 1.

**Table 1.** Data Sources and Methodologies for the Data Analysis in This Study.

| Case Study | Analysis Unit | Data Sources | Methodologies for Data Analysis |
|---|---|---|---|
| IPE | Structural Social Capital | The website of IPE<br>45 technical reports and project reports produced by IPE<br>3 annual reports of IPE<br>7 archival documents<br>2 interviews with IPE representatives<br>Academic publications | Theoretically informed qualitative content analysis<br>Quantitative social network analysis via Python package NetworkX |
| | Cognitive Social Capital | 2 interviews with IPE representatives<br>3 annual reports of IPE<br>Academic publications | Theoretically informed qualitative content analysis |
| Green Hunan | Structural Social Capital | The website of Riverwatcher<br>The mobile app of Riverwatcher<br>5 annual reports of Green Hunan<br>2 annual reports of Riverwatcher<br>10 monthly reports of Riverwatcher<br>1 interview with a Green Hunan representative | Theoretically informed qualitative content analysis<br>Quantitative data analysis via equation solving |
| | Cognitive Social Capital | 22 archival documents<br>Academic publications | Theoretically-informed qualitative content analysis |

At the core of structural social capital are the structural characteristics of networks, such as the strength of social relations, which can be objectively measured and better described through quantitative methods. The Python package NetworkX offers a well-tested, well-documented, effective, and easily accessible tool to create and study the structure, dynamics, and functions of social networks [61]. This package was used to quantitatively analyze the structural aspect of IPE's business network. As for the case study of Green Hunan, an equation was proposed to calculate and explain the strength of social relationships in the Riverwatcher Action Network.

Qualitative methods are well-suited for exploring questions about interactions, attitudes, values, motivations, and perspectives [62]. Therefore, qualitative research provides a suitable approach to make sense of not only cognitive social capital, but also key aspects of structural social capital, including the presence and number of network members and social interactions among them. This study employed content analysis to classify, organize, and interpret the qualitative data. Content analysis can be defined as " . . . a research method for the subjective interpretation of the content of text data through the systematic classification process of coding and identifying themes or patterns." [63] (p. 1278). Social capital theory—in the previously defined theoretical proposition—informs and guides the coding process that is a crucial part of qualitative research in this study, as illustrated in Table 2. Three core themes were identified to find patterns of actions and consistencies in the network of relationships embedded in the data [63]. Multiple codes of each theme were created to systematically link the empirical data to the research questions and the theoretical foundation. The first theme involves social interactions and relations among network members. Five codes are clustered

under this major theme: IPE–MNCs, MNCs–polluting suppliers, polluting suppliers–IPE; Green Hunan–volunteers, and volunteers–volunteers. The shared concern and objective of network members constitutes the second theme. Within this thematic category, this study developed a set of codes: MNCs' concerns about environmental management in the supply chain; polluting suppliers' motivation to take corrective actions; the biospheric value orientations of volunteers in the Riverwatcher Action Network; volunteers' social–altruistic attitudes; volunteers' egoistic motivations. Moreover, the third theme concerns the shared culture of networks. Two codes exist under this category: the rules and norms in IPE's green supply chain network and the manner of participation of the volunteers in Riverwatcher Action Network. Finally, using the results of the data conceptualization process, coding, and interpretation, the explanation of how structural and cognitive social capital were created based on big data adoption in both case studies was devised. These empirically-based findings are presented in the section below.

**Table 2.** Themes, Codes, and Descriptions Used in This Study.

| Theme | Code # | Code | Description |
|---|---|---|---|
| Social interactions and relations | R1 | IPE–MNCs | Communication and engagement between IPE and MNCs about the identification and resolution of pollution problems in the supply chain |
| | R2 | MNCs–polluting suppliers | MNCs' interacting with polluting suppliers |
| | R3 | Polluting suppliers–IPE | Manner of engagement of polluting suppliers with IPE to address pollution problems |
| | R4 | Green Hunan–volunteers | Volunteers' participation in Green Hunan's Riverwatcher Action Network |
| | R5 | Volunteers–volunteers | Interactions in volunteers' monitoring groups |
| Shared concern and objective | O1 | MNCs' motivation | MNCs' concerns about corporation reputation and profitability |
| | O2 | Polluting suppliers' motivation | Polluting suppliers' concerns about losing business |
| | O3 | Volunteers' social–altruistic attitudes | Volunteers' concerns about damage caused by water pollution in communities |
| | O4 | Volunteers' biospheric value orientations | Volunteers' concerns about the river itself |
| | O5 | Volunteers' egoistic motivations | Volunteers' concerns about their own interests affected by water pollution |
| Shared culture | C1 | Greening the supply Chain | Rules and norms in IPE's green supply chain network |
| | C2 | Monitoring water pollution | Manner of f volunteer participation in the Riverwatcher Action Network |

## 4. Empirical Findings

*4.1. The Institute of Public and Environmental (IPE) Affairs*

### 4.1.1. Green Supply Chain Management

IPE uses technologies to capture environmental data from multiple sources, manage and integrate the data in a systematic and informative way, and store and visualize the data in its environmental map database, which is the cornerstone of IPE's big data adoption. China has long been seen as

an "information-poor" country, and citizens have limited access to environmental information [64]. However, recent years have witnessed remarkable progress in information disclosure and environmental transparency. The major breakthrough occurred in 2008, when the Regulation of the People's Republic of China on the Disclosure of Government Information and the Measures for the Disclosure of Environmental Information took full effect. The two landmark laws require all levels of the Chinese government and all enterprises to disclose a wide spectrum of information—ranging from environmental quality to pollution discharge and emission, to environmental management and inspection—leading to the vast and rapidly increasing volume of environmental data in the country. By virtue of the comprehensive scope of official environmental data and the lack of an alternative source, the governmental data are widely viewed as the major source of reliable environmental information in this authoritarian setting [65].

Against this background, IPE collects large-scale data from a multitude of official sources, including 31 environmental protection agencies at the provincial levels, 338 environmental protection agencies at the city and county levels, other related agencies at all levels such as Water Resources Department and the Department of Land and Resources, and official media reports. The real-time monitoring by key industrial polluters, which is mandated by law and disclosed via official channels, is also an important source of IPE's data. After extracting the necessary data from the large, complex, heterogeneous, and unstructured datasets, IPE adopts a series of rules to transform the extracted data into standard and well-interpretable formats. The crucial phase of big data analytics is to import the transformed data into the target storage infrastructure and use maps to visualize the data [66]. The data acquisition, extraction, integration, storage, and visualization lead to the creation of an environmental map database that provides useful information and displays a unified view of the data in a clear and easily understandable format as well as on a real-time or near real-time basis [67–69].

Big data are typically defined by the 3Vs: volume, velocity, and variety [70]. Volume concerns the immense size of the data as they either consume a huge amount of storage or consist of a large number of records. Indeed, volume represents big data's primary attribute [71]. Velocity refers to the frequency or the speed of data generation and delivery, and indicates that data are created and processed rapidly and in a timely manner [72]. Variety concerns the various forms of data, namely, structured data (data from relational databases) and unstructured data that cannot be processed using traditional analytical tools, such as text, webpage, images, video, and audio data [73,74]. IPE's environmental map database is described and understood in terms of the defining attributes of the 3Vs, as illustrated in Table 3.

**Table 3.** 3Vs of The Institute of Public and Environmental Affairs (IPE's) Environmental Map Database.

| Type of Data | Volume | Velocity | Variety |
|---|---|---|---|
| Air Quality | Data on air quality index (AQI), $PM_{2.5}$, $PM_{10}$, $SO_2$, $NO_2$, $O_3$, and CO from 4092 monitoring stations nationwide | Real-time data every hour | Web data, text documents |
| Water Quality | Data on water quality grades from 23,344 monitoring stations nationwide | Real-time data every four hours; real-time weekly or monthly data | Web data, text documents |
| Air Pollution Emission | Data on air pollutant concentrations (NOx, $SO_2$, and dust) from 7917 primary sources of air pollution | Real-time data every hour | Web data, text documents |
| Water Pollution Discharge | Data on water pollutant concentrations (COD, pH, NH3-N) from 15,646 primary sources of water pollution | Real-time data every hour | Web data, text documents |
| Environmental Violation Records of Enterprises | Over 1,545,000 violation records of more than 1,052,700 polluting enterprises (date, type, cause, penalty, inspectors of environmental violations, and correction and verification information) | Updated per day | Web data, text documents images |

The environmental map database is characterized by the large-scale and high-velocity data of the environmental violation records of more than one million enterprises' as shown in Figure 1. Enterprises have been major contributors to China's environmental degradation. IPE has transformed these big data of the enterprises into a supply chain management tool to motivate the business community to increase environmental performance and enhance their accountabilities. Central to this big data-based tool are the MNCs that use the data to monitor and supervise the environmental practices of its suppliers in China. A MNC often relies on hundreds or even thousands of Chinese suppliers; in fact, China has long been called the "world's factory." MNCs are typically concerned about being compliant with corporate environmental responsibilities and maintaining a good reputation. Thus, many of them have made environmental supply chain management a part of their respective business strategies [75]. This creates incentives for them to collaborate with IPE and push the polluting suppliers to correct their behaviors. A MNC is often the primary buyer of a local enterprises' products, which gives MNCs a powerful position to exert market pressure on their suppliers [76]. Enterprises that take corrective measures and comply with environmental regulations are allowed to continue contracting with the MNCs. However, enterprises that do not take corrective action and do not abide by the regulations are sanctioned through the loss of contracts and revenues. Thus, communications and interactions among IPE, brand-sensitive MNCs, and suppliers led to the formation of a relationship network, which in turn leads to the creation of social capital.

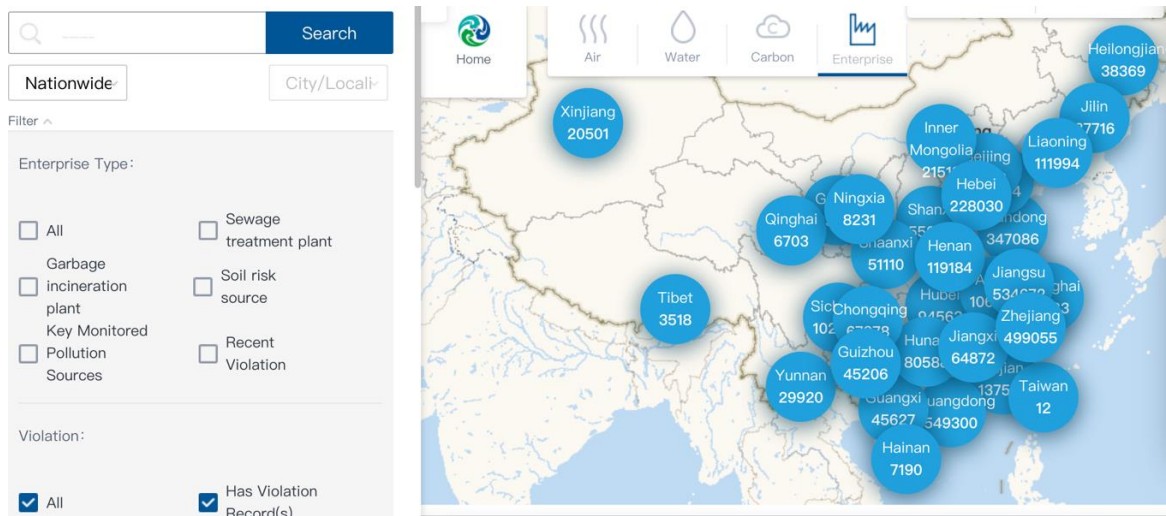

**Figure 1.** IPE's environmental map database displays information on enterprises' environmental violation records in every province (blue circles), and is updated every day [77]. The data of all Chinese provinces can be acquired by zooming in.

4.1.2. Structural Social Capital: IPE–Brand-Sensitive Multinational Corporations (MNCs)–Chinese Suppliers' Network

Using Python package NetworkX 2.4 (2019), this study presented a typical structure of a social network showing where IPE, MNCs, and suppliers were located and how they interacted with one another between October 2015 and September 2016, as shown in Figure 2 [78]. Social capital is embodied in and derived from four types of social interaction ties, as represented by the lines in four different colors. This study analyzed the network positions and the interconnectivity of the three actors, with an emphasis on the structural characteristics of the network. First and foremost, the relations between IPE and the MNCs (in black) played a central role in initiating and sustaining the network. Relying on the database, IPE identified the polluting enterprises believed to be the suppliers of some large and influential MNCs in a variety of sectors such as IT, textile, and leather. Then, IPE actively reached out and engaged with over 50 MNCs using different means, including letters, emails, phone calls, teleconferences, and face-to-face meetings, to make them aware about their suppliers'

violations and persuade them to take action. Moreover, recently, a growing number of MNCs have begun proactively and regularly—on a quarterly, monthly, or more frequent basis—to search through IPE's environmental map database to check whether a supplier violates environmental regulations or standards. It is important to consider the context here. China's environmental information had been widely scattered across multiple sources, which made it very difficult, if not impossible, for MNCs to identify and locate environmental problems and the violating enterprises in their supply chains, thus working as a disincentive for MNCs to take action. The emergence of IPE's environmental map database thus represents a watershed change and provides MNCs with a useful tool to manage their supply chains.

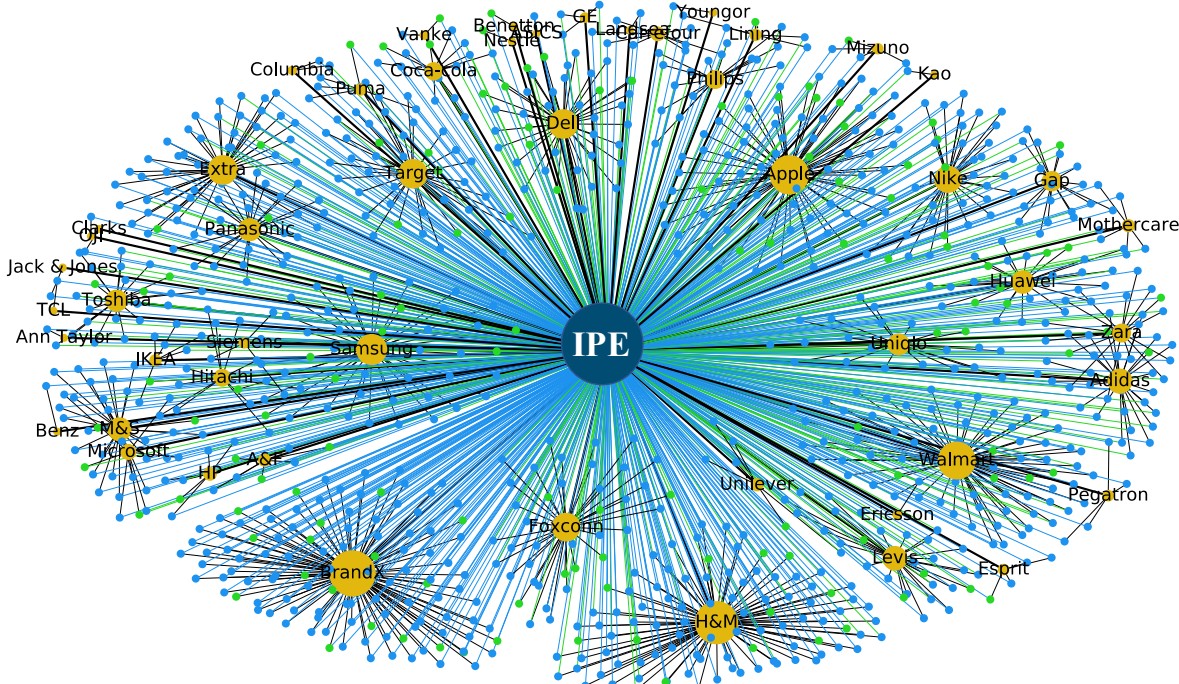

**Figure 2.** The social network among IPE, Brand-Sensitive MNCs (yellow circles), Chinese Suppliers at the 1st level (blue circles), Chinese Suppliers at the 2st level (green circles). Four types of social interaction ties are represented by lines in different colors (black: IPE-MNCs; grey: MNCs-suppliers; blue and green lines denote suppliers-IPE interactions at the 1st and 2nd levels, respectively). This network is plotted with the Python package NetworkX [79]. Some MNCs did not want to disclose their names. They were aggregated into one group called "BrandX" in the network in Figure 2. Data and codes can be found in Supplementary Materials '1-Data_Figure2.zip'.

Due to the communication and interaction between the MNCs and IPE, the second social relation ties (in grey; Figure 2) emerged between the MNCs and the polluting suppliers. Concerned about their corporate reputation and image, the MNCs were motivated to deal with the polluters and pushed them to take corrective measures. They required them to communicate and engage with IPE about information such as explaining their violations, providing corrective action plans, and ensuring that the corrections were verified by IPE. Each MNC in the network is represented by a yellow circle whose size is determined by the number of polluting suppliers that actually interacted with IPE to address their violations. For the sake of identification, the MNCs' names are written in the circles. The size of each circle indicates the strength of the social interaction ties between IPE and the MNCs. The bigger the red circle, the greater the number of polluting suppliers taking positive steps to behave in environmentally responsible ways, and the stronger the ties. As illustrated in Figure 2, Walmart, Apple, H&M, Samsung, Dell, and Target are in the top tier in terms of how many suppliers they effectively pushed to make

pro-environmental behavior changes. Moreover, this result indicates that they interacted with IPE more intensively and frequently, and thus, they stand out with the strongest ties.

To avoid losing contracts with the chief buyers, the polluting suppliers (described by the blue circles, Figure 2) reached out and worked with IPE on two levels, leading to two patterns of social interaction ties between them. At the first level, approximately 730 polluting suppliers communicated with IPE and provided documents and information in order to demonstrate how they dealt seriously with such violations. This included an explanation of the violations, corrective action plans, official reviews and approvals of their corrections and compliance, and follow-up monitoring documentation. This pattern of social interaction ties is illustrated by the blue lines. Moreover, considering that environmental evaluation and verification is highly technical in nature, IPE established a Green Choice Audit (GCA) system to carry out a rigorous audit of an enterprise's environmental practices by an accredited third-party auditor and IPE. This arrangement created a robust verification process and led to a deeper relationship. During the period between October 2015 and September 2016, 272 polluting suppliers went through the GCA and got their corrections verified [80]. The green lines are used to indicate this level of social interaction ties between the polluting suppliers and IPE. Due to the establishment and implementation of the GCA system, the violation records of the polluting suppliers who later made amends could be removed from the "violator" blacklists in the database [81]. Moreover, IPE arranged all the related documents and information concerning the above-mentioned ties at two levels in the database. The client MNCs, therefore, could check the database to monitor the response actions of their suppliers. It is worth noting that the intention of the interplay between IPE and the polluting suppliers was to start a conversation, share information, offer assistance, and seek a solution to the suppliers' problems. IPE was oriented toward helping enterprises create environmental action plans and take corrective measures, rather than penalizing them or shutting them down.

The social network among IPE, the MNCs, and the suppliers experienced steady expansion and growth, with an increasing number of enterprises getting involved. Walmart, Nike, GE, Extra, and Unilever are considered as pioneers that started cooperating with IPE in the late 2000s [79]. IPE has forged enduring and cooperative relationships with numerous MNCs many of which are shown in Figure 2. Moreover, new MNCs, such as New Balance, C&A, Inditex, and V&F, continue to emerge and become active actors [82]. In 2018, the number of the polluting suppliers pushed by the MNCs to engage with IPE for addressing their violations climbed to 2458—over three times the number in 2016—and even more tellingly, the number of the polluting suppliers that went through the GCA system increased by nearly five times to 1206 [83]. The surge in these numbers can be largely attributed to the long-term cross-sector cooperation between the MNCs and IPE, suggesting that their social interaction ties have grown stronger and more effective. Structural social capital was thus created, sustained, and enhanced.

### 4.1.3. Cognitive Social Capital: Greening the Supply Chain and Business-Environmental Non-Government Organizations (ENGO) Partnerships

The creation of structural social capital has been accompanied by shared norms and understanding concerning the big data-induced environmental supply chain management. The multifaceted relations among IPE, MNCs, and suppliers are centered around one shared goal, namely greening the supply chain, even though these actors have different motivations. Greening the supply chain can be described as a set of approaches and practices for managing and achieving effective coordination and collaboration between organizations to minimize negative environmental effects in the supply chain [84,85]. Incorporating environmental considerations into procurement and purchasing is one key phase of greening the supply chain [86]. Enterprises can be driven by a series of motivations to manage their supply chains in a pro-environmental manner, including regulatory pressures, positive corporate reputation, stakeholder pressures, resource savings, cost reduction, and higher business profitability [87,88]. In this case, the primary driver for the famous MNCs included in this study lies in their concern about branding and corporate reputation [77,80,89]. Herein, corporate reputation

refers to a consumers' overall evaluation of a company's strength or capacity to handle environmental issues, which has great influence on the company's performance and consumers' behaviors [90]. MNCs typically make commitments regarding environmental protection. It is now widely accepted that assuming environmental responsibilities is an integral part of business ethics. MNCs' active involvement in managing and greening supply chains can demonstrate their actions and efforts to fulfil their commitments, which also ensures meeting consumer expectations. Doing so can help build, protect, or improve their reputation; failure to do so can lead to reputational risk [91]. For example, in 2010, IPE repeatedly reached out to Apple, IBM, and Canon about heavy metal pollution caused by illegal discharge of the enterprises believed to be their Chinese suppliers. Nevertheless, these top MNCs chose not to respond. As a result, over 1000 consumers within China and abroad wrote letters to these MNCs to express their concerns about pollution problems in their supply chains and demanded a response [92,93]. Due to the heightened consumer pressures and reputation concerns, the MNCs finally started to communicate with IPE and worked together to address the problems. In addition to MNCs' reputational concerns, regulatory pressure and environmental liability have recently become increasingly important driving forces that motivate MNCs actions. As China's environmental enforcement has become stricter in recent years, the violating suppliers that do not take corrective measures are likely to be ordered to suspend production [89]. As a result, MNCs will suffer from supply problems, which would negatively affect their profitability. Moreover, the violating suppliers, who face the consequences of losing big clients, need to align their behaviors with the MNCs' goals of greening the supply chains. This shared objective is also in line with IPE's mission to unleash the power of big data to involve the business community in enhancing environmental governance.

Environmental regulators in China have primarily relied on the top-down command-and-control approaches with regard to an enterprise's environmental problems. However, due to the big data-induced management tool used by IPE, the MNCs and the suppliers represent a much more horizontal structure which emphasizes conversation, negotiations, assistance, and partnership. This new set of rules and norms contributes to greening supply chains. First, IPE and the MNCs rely on environmental data to initiate and legitimize their activities. The environmental map database is exclusively built using governmental data. The government is regarded as an authoritative and reliable source of environmental data in China [64]. Therefore, relying on these environmental data offers legitimacy and political protection to all actors, particularly to IPE, as NGOs often face problems associated with non-recognition, lack of legitimacy, and low influence in the country. Their data-driven activities provide credible evidence of the suppliers' violations, thus reassuring the suppliers that they are not unfairly targeted, and that it is their responsibility to solve the violations. Second, environmental officials seek regulatory authorities over polluting enterprises to enforce laws and regulations. However, in the above-mentioned network, the MNCs play the role of the defacto environmental enforcer by exploiting their market positions and exerting economic leverage on their suppliers. The major difference between them is that the former exercise mandatory requirements for the suppliers to change their behaviors, while the latter encourage voluntary corrective action. Last, but not least, the multifaceted relationships among IPE, the MNCs, and the suppliers are intended to build communication and engagement across sectors and to promote information sharing and non-confrontational discussion. ENGOs have comparative advantages in providing environmental solutions and producing ideas and giving advice to address environmental issues [94]. In practice, IPE often offers ecological expertise and assistance to help, rather than to penalize the violating suppliers or shut them down. The data-driven, conversation-based, and assistance-oriented approach presents a new way of acting and interacting to address environmental problems and building ENGO–business partnerships.

### 4.2. Green Hunan

#### 4.2.1. A Mobile Data-Based Riverwatcher Action Network

In 2011, Green Hunan launched the Riverwatcher Action Network initiative to mobilize and engage ordinary citizens in water pollution monitoring, thereby building a network of social relations among Green Hunan and volunteers. Many volunteers were recruited, trained, and linked to each other. Eight monitoring groups were created to collectively maintain a watchful eye on the Xiang River and two industrial districts in Hunan province [95]. They set up numerous specific monitoring sites, monitored the water quality of the river and the discharges of the enterprises on a regular basis, identified pollution problems in a timely manner, and promptly took photos or videos of pollution and made notes (e.g., describing the color and odor of the water) and individual comments [96]. The volunteers relied on the human senses of sight and smell, as well as water quality testing equipment, such as pH test strips, to identify water pollution incidents. They often took water samples at discharge points and sent them to Green Hunan for further water quality testing [97]. Besides these major measures, the volunteers also conducted field investigations and interviews with local communities to better understand and monitor the occurrences and sources of water pollution [98]. The information they collected in this process constituted a new type of environmental data. The next critical step was to share the data, in a manner as widely and timely as possible, over the internet, social media, mass media, and official pollution reporting channels [19,99].

The creation and sharing of such data immediately drew the attention of both the public and the government, created bottom-up pressure on the polluting enterprises and environmental regulators, which helped facilitate the identification and resolution of the problems. For example, on 11 September, 2011, a volunteer group of Green Hunan found that a large amount of red-colored wastewater effluent was released from a local industrial district in Xiangtan city [95]. A volunteer pulled out his cellphone and immediately took a photo (shown in Figure 3). He posted the photo on Microblogging, China's equivalent of Twitter, alongside his comment noting "Our monitoring team is in a pollution-intensive district called Zhubugang. Red effluent is being discharged directly into the Xiang River. It is outrageous." Within hours, the message quickly went viral, leading thousands of Microblogging users to repost it and comment. The photo was also widely displayed on major news and media websites. An official of the local environmental protection bureau soon contacted the volunteer to ask about the location of the pollution incident and conducted a prompt investigation. As a result, the company as responsible for the discharge of the red effluent was ordered to close, and its Chief executive officer was also ordered to publicly apologize on the Xiangtan Daily News.

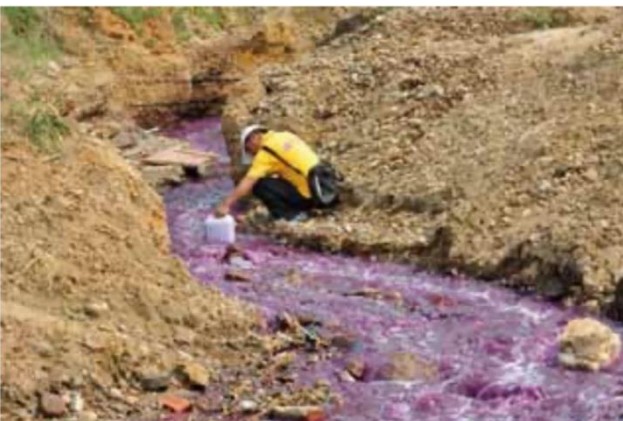

**Figure 3.** A photo was taken by a volunteer of Green Hunan and posted on microblogging on 11 September, 2011, showing red wastewater effluent was discharged directly into the Xiang River [92].

Green Hunan created a useful mobile data-based model for public participation in water protection and pollution monitoring, leading to a rise in the number of volunteers and the rapid expansion of the Riverwatcher Action Network. In 2015, four years after the network was launched, 30 monitoring groups, consisting of 271 key volunteers, were formed [19]. The network has grown and extended the scope of its monitoring far beyond the Xiang River it was initially designed to cover; it now focuses on three more rivers (the Li, Yuan, and Zi Rivers) across 52 cities, counties, and districts of Hunan province, as illustrated in Figure 4 [19]. Green Hunan and its volunteers worked together at a large number of monitoring sites scattered throughout Hunan province. All of the four major rivers of the province were, therefore, largely under the watch of volunteers who carried out monitoring activities in a regular, timely, persistent, and site-specific manner. In 2015 alone, volunteers took a total of 9047 photos of water pollution incidents and created 2078 monitoring records. They shared such environmental data through the internet and social media on a real-time or near real-time basis, which contributed to addressing hundreds of pollution problems [19]. Since 2017, Green Hunan has been developing and promoting the Riverwatcher Action Network in many other areas, moving beyond the Hunan province to cover the entire country. Social capital resides in social networks and social relations [35]. Green Hunan and its volunteers engage in interactions, undertake coordination, and ensure cooperation to identify and solve the water pollution problems they commonly face, thereby building and strengthening social capital.

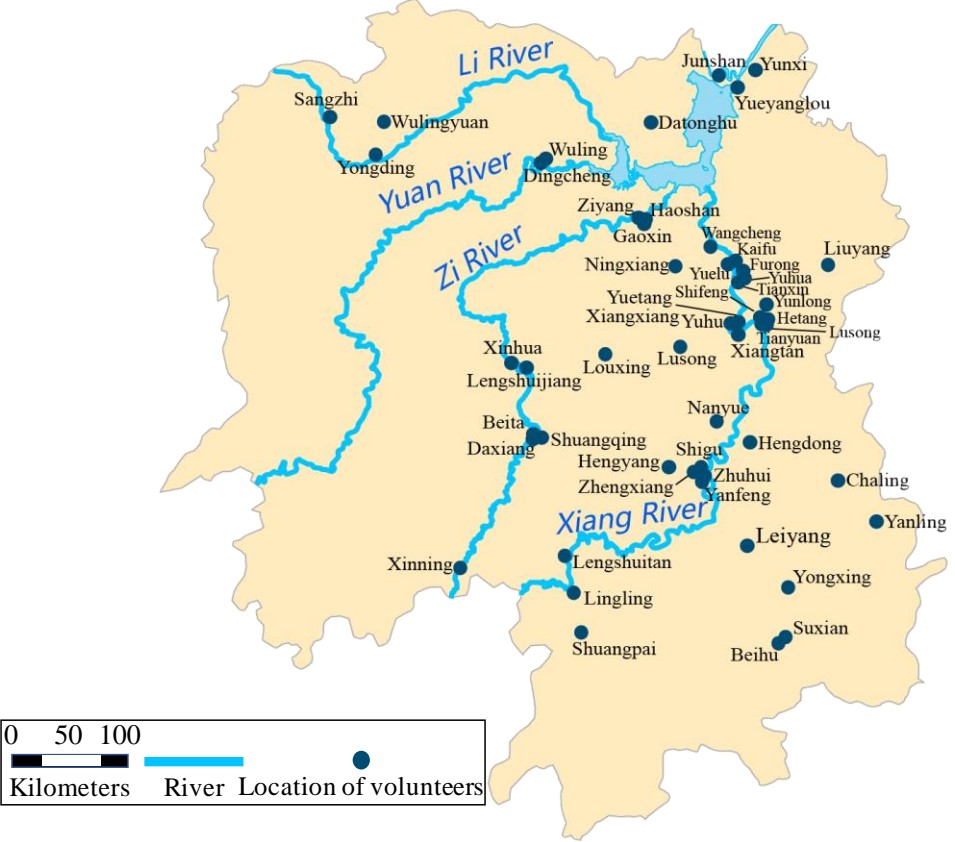

**Figure 4.** In 2015, Green Hunan's Riverwatcher Action Network of 30 monitoring groups consisted of 271 key volunteers that kept watchful eyes on the water quality of all the four major rivers of the Hunan province across 52 cities, counties, and districts of the Hunan province. Data can be found in Supplementary Materials '2-Data_Figure4.zip'.

### 4.2.2. Mobile App-Based Structural Social Capital

The 2010s have seen the dramatic rise in the use of mobile technologies and the huge surge in the number of mobile phone users in China. The ubiquity of mobile phones indicates the presence

of a vast untapped pool of potential volunteers for ENGOs. Against this backdrop, Green Hunan invented and launched a mobile application called Xunhebao in April 2019 [99,100]. The function of the app is four-fold. The first is to make it easy for volunteers to create and share environmental data regarding their water pollution monitoring activities, and therefore to network more people nationwide. The app not only helps volunteers to easily take a photo of a pollution incident and tag it with the location the photo was taken at, but also helps them to better describe the situation and make comments/observations. It also makes it much easier for volunteers to systematically record their monitoring activities and manage their data. Second, the app facilitates information exchange, communication, interactions, networking, and cooperation among volunteers. Users can easily team up with other users to act collectively and take care of water bodies in a specific geographical area, with the team size of hundreds of people. The app allows relations-building, such as via a ranking system for team members. Third, the app enables users to quickly and easily share the data and post it to WeChat, China's most popular and influential social media platform, Microblogging, and other various online platforms. In order to facilitate addressing water pollution problems, the app provides a service that helps volunteers share the data with the government in a timely manner. More tellingly, each week, Green Hunan extracts the data from all users and integrates it into weekly reports on water quality and pollution of specific rivers, valleys, or geographical areas, and then offers the reports to the corresponding governmental agencies. Fourth, Green Hunan has recognized the importance of big data in contributing to environmental governance. It aims to use the app to better collect, accumulate, manage, and integrate environmental data derived from this large-scale civic engagement in water protection.

As at 30 November, 2019, eight months after the use of the app began, the monitoring groups of interacting volunteers reached 1033 in number, consisting of more than 110,000 users throughout the country [101]. Volunteers have conducted monitoring activities, taken photos, and produced and publicized data at over 59,000 sites between April 2019 and November 2019 [98]. In terms of the aforementioned 3Vs that define the characteristics of big data, the data created and shared by the volunteers in the Riverwatcher Action Network constitute a new source of environmental big data. A huge volume of data originates from hundreds of thousands of volunteers that live across most of China, covering 22 provinces and 7 provincial-level municipalities and autonomous regions, as shown in Figure 5 [101,102]. As the number of users keeps rising, the volume of the data is expected to grow exponentially. To capture the true and real-time situation of the water bodies, the volunteers take on-site photos and share them publicly in a timely fashion. Therefore, the data are generated and delivered at high velocity. Furthermore, the data takes three unstructured forms, namely images, text, and video, exhibiting the variety of big data in a context where the main types of environmental data typically consist of structured data and unstructured webpage and text documents.

The creation and adoption of these new big data make it possible for ENGOs and ordinary citizens to build networks of social relationships from which extensive social structural capital is derived. For the sake of clear and concise exposition, this study emphasizes the aggregate-level attributes of the network in this analysis. At the aggregate level, the strength of social relationships can be measured by the combination of two factors at the provincial level: $F_1$ is the number of monitoring groups of volunteers that adopt the app, and $F_2$ is the number of monitoring sites where environmental data have been created and shared by the volunteers, with both factors being assigned a weighting ($W_1$ and $W_2$) of 50%. The strength of the social relationships is thus expressed as follows [99,101,102],

$$S = W_1F_1 + W_2F_2 \tag{1}$$

As indicated in Figure 5, while structural capital has been developed across most of China, it is higher in the Hunan, Anhui, Guangdong, and Zhejiang provinces. The lower levels of structural capital are found in the Gansu, Qinghai, Ningxia, Inner Mongolia, Jilin, and Hainan provinces. Structural capital is absent in the two provinces of Tibet and Xinjiang.

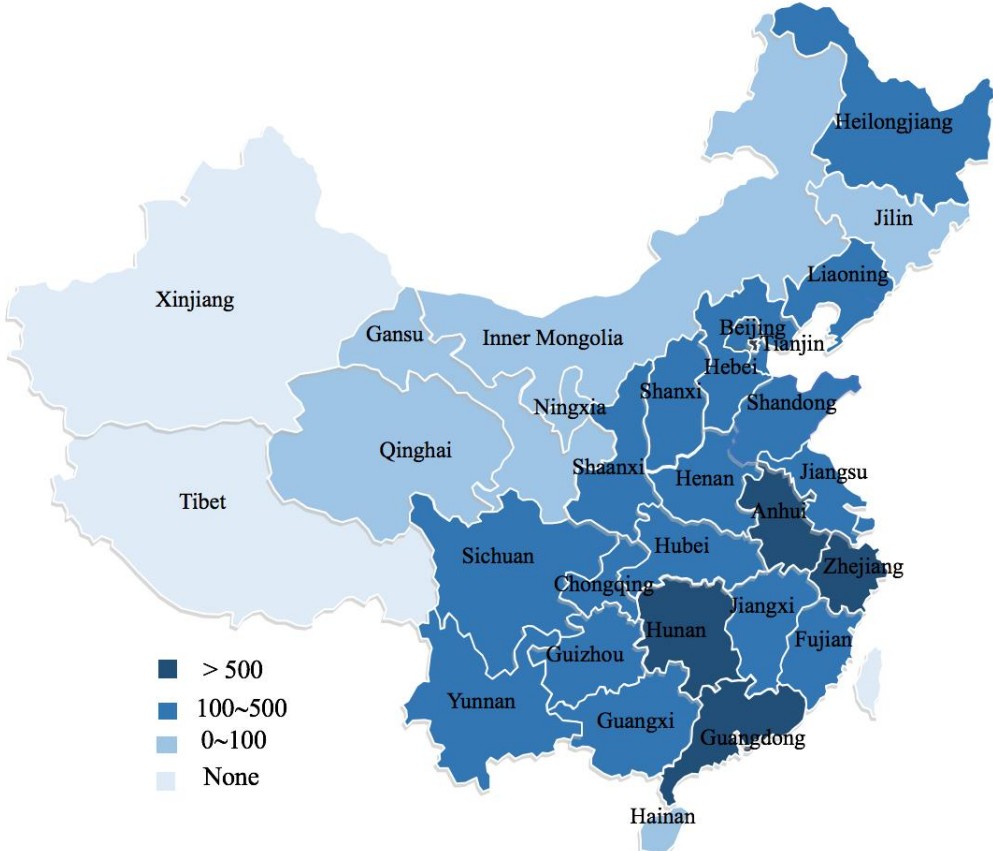

**Figure 5.** The maps show the strength of social relationships in the Riverwatcher Action Network at the provincial level, which is calculated with the data between April and November 2019. Data can be found in Supplementary Materials '3-Data_Figure 5.xlsx'.

### 4.2.3. Cognitive Social Capital: Three-Dimensional Value Orientations Towards Water Pollution

The classic Value–Belief–Norm (VBN) theory of pro-environmental behavior proposes that egoistic, social-altruistic, and biospheric value orientations causally influence how people cognitively structure beliefs regarding negative environmental consequences [103–108]. According to the VBN model, the volunteers' engagement in the Riverwatcher Action Network is motivated by value-based concern in three dimensions: Their biospheric concerns about rivers and other water bodies, their social–altruistic concerns about other people, and their egoistic concerns about themselves. First and foremost, the volunteers' emotional connection and bonding to rivers and other water bodies plays a significant role. In the early 2010s, the water quality of 39% of China's major rivers was classified as Grade IV or worse; in other words, the water was unfit for human contact [109]. Even worse, the water, which was too polluted to be touched by humans, accounted for approximately 58% of the country's major lakes. Against this backdrop, an increasing number of people became volunteers and got involved in the network. They generally call the rivers they voluntarily monitor and defend as the "Mother River" and believe that people's lives, from generation to another, depend on the water bodies. A volunteer noted that "Xiang River is our 'Mother River' that not only nurtures us but is our spiritual harbor as well." Many of the volunteers have witnessed the change from clean and safe water bodies to the current conditions fraught with pollutants, which arouses a sense of moral responsibility [110]. Another volunteer shared the view that, "After walking along the Liuyang River, our Mother River, I see that it was hurt badly by the degradation of water quality. It is our responsibility to care for it. Protecting the river is a long and difficult road. But if more people get involved, we will have limitless strength to restore it" [111].

Moreover, ordinary citizens feel the impact of water quality degradation on a daily basis. Over 90% of China's groundwater has been contaminated, which poses a serious risk to the safety of drinking water supplies and public health [112,113]. It is estimated that water pollution-related diseases cause sicknesses in 190 million people and kill 60,000 people in China every year [114]. The number of "cancer villages" widely believed to be caused by water pollution has grown to approximately 600 across the country [115]. Unsurprisingly, water pollution ranks among the top 3 perceived environmental threats in the minds of the Chinese public. In a survey, 71.8% of respondents perceived water pollution as an active threat [116]. It is in this context that the volunteers are driven by their concerns about the consequences of water pollution for people in the community and themselves. Their social–altruistic orientations can be illustrated by the feelings expressed by two other volunteers. One dedicated volunteer pointed out, "I want people in our community to have clean water to drink and clean air to breathe, which is the only motivation for me to keep monitoring and reporting pollution problems." [117]. This is echoed by another volunteer, who said, "My regular monitoring site is an entrance to a local drinking water company. This place is very important because it is related to the safety of drinking water for hundreds of thousands of people in our city" [118]. The volunteers with egoistic orientations are concerned that water pollution inflicts damage or risk on themselves. "We are victims. Our lives are affected by the pollution in the rivers near our neighborhood. We have to do something with it" [119]. This notion represents a simple and straightforward reason to become volunteers. Overall, Green Hunan's volunteers in the Riverwatcher Action Network share concerns covering three correlated value orientations over water pollution problems, leading to their common objective of working together to address these problems.

In tandem with the shared concerns and objective, the volunteers have a shared perspective on how to act collectively under Green Hunan's Riverwatcher Action Network. First, they believe that local people are key to solving local environmental problems. Therefore, the volunteers are usually familiar with local environmental and social conditions and focus on monitoring the condition of their local waters, such as the rivers near their homes or workplaces. Second, to ensure that the water pollution problems are addressed, the first step involves identifying the problems in a timely fashion and creating credible monitoring data. The volunteers play the role of monitors and reporters on the ground by regularly identifying otherwise undiscovered pollution incidents and unmeasured or uncollected environmental data. Third, they rely on ICTs, including mobile applications and social media, to spread the data, to make the public and the government aware about the occurrences of water pollution on a timely basis [99]. Thus, bottom-up pressure is created and applied on both the polluting companies and environmental regulators, driving them to take their environmental responsibilities seriously and to hold them accountable. At the core, their monitoring activities are intended to raise attention and motivate action.

## 5. Discussion

Drawing on social capital theory and evidence from empirical case studies, this research shows that the adoption of big data in ENGOs helps build bridging and bonding green social capital in China. IPE and Green Hunan stand out as excellent examples in that they take good advantage of big data to address environmental problems. The essence of big data adoption by IPE lies in its capacity to search, aggregate, and cross-reference large environmental datasets. IPE's environmental map database has induced the development of bridging social capital among IPE, brand-sensitive MNCs, and suppliers, the latter two being heterogeneous groups in terms of their socio-economic status. These three network members interact with one another and constitute a social network that consists of four different types of social relations for a shared goal: managing MNCs' supply chains in a pro-environmental manner. A set of shared principles—including reliance on environmental data, exerting market force to prompt voluntary actions, and focusing on communication and engagement—have structured appropriate behaviors and perceptions of the network actors and facilitated their coordination and cooperation. These rules, in conjunction with the presence of the shared goal, comprise cognitive social capital, a

core component of social capital. Furthermore, since 2011, Green Hunan has developed and expanded the Riverwatcher Action Network. Numerous volunteers, who share value-based concerns and similar attitudes over water pollution, act collectively to produce and deliver mobile-based environmental data. The communication and social ties among them represent a horizontal structure and lead to the generation of bonding green social capital. Under the leadership of Green Hunan, the volunteers follow a set of common rules and norms to govern their actions. For example, local people are well aware about, and are the most affected by local pollution problems; they are thus key to solving these problems. In other words, as explained in considerable detail in Section 4, the use of big data enhances ENGOs' ability to facilitate collective actions, mobilize environmental activism on the ground, and promote stakeholder engagement in environmental issues.

The advent of environmental big data opens up new avenues for ENGOs to play the role of a change agent in facilitating a sustainability transition. Today, Chinese ENGOs are striving to work with multiple actors within and across different sectors to help solve environmental challenges that no single actor can solve. Big data-based activities provide a new way of engaging these stakeholders, including the general public, the business community, and the government, to act and coordinate more actively and effectively. The heart of this change agency is associated with creating and using environmental data on a large scale and in a timely fashion. A myriad of cellphones throughout the country have become a mobile environmental micro-monitoring station. ENGOs, which act as catalysts for change, mobilize and encourage ordinary citizens to create data independently and participate actively in environmental protection. While one citizen's contribution may be small, the cumulative effects of large-scale pubic participation can make a big difference. Moreover, the ENGOs collect, analyze, and extract knowledge and insights from a great volume of data to serve as change agents. While China's official, high-volume environmental inspection records are publicly available to all, only certain ENGOs have been able to extract the valuable information on the violations of MNCs' suppliers on a timely basis. Further, these ENGOs have played a central role in initiating, managing, and implementing pro-environmental changes in MNCs' supply chains.

Big data usage by the ENGOs and its effects on generating green social capital contribute to China's environmental governance in many respects. Five key elements of governance exist: An array of institutions and actors that are drawn not only from, but also beyond the government; the blurring boundaries and responsibilities with regard to addressing social problems; power dependence; the autonomous self-governing network of the actors; the capacity to get things done, which does not rely on the power of the government or use its hierarchical authority [120,121]. The two big data-based activities highlighted in this paper—green supply chain management and the Riverwatcher Action Network—are characterized by the critical role played by non-state actors. Environmental responsibilities have been taken up by non-state actors, including ENGOs, the business community, and the general public. The government is no longer solely responsible for environmental problems. The non-state actors are committed to shared concerns and dependent on one another to address such collective-action problems as water pollution issues that cannot be solved by any single actor. Thus, they blur the boundaries of the public, private and voluntary sectors. What is striking is that the use of big data has significantly increased ENGOs' capacity to mobilize stakeholders, bridge multiple actors for meaningful participation, leverage each other's resources and capabilities, and ultimately bring about an interactive governance process with power dependence. Moreover, governance concerns the autonomous self-governing networks of actors, as illustrated by the IPE–MNCs–suppliers' network. It was created as an informal structure and its processes for interaction and communication lacked an all-encompassing structure of command. Environmental violations in MNCs' supply chains can be addressed by exerting financial pressure, building engagement and dialogue, and providing assistance without resorting to the governmental authority. Overall, an important insight into the governance benefits of big data can be gained by considering how the application of big data is implicated in the processes of building social capital.

The authors drew on the literature on big data, social capital, and ENGOs for this research which, in turn, made theoretical contributions to enrich these three research fields. First, existing literature on big data primarily takes the perspective of technology, and focuses on big data analytics and application, such as state-of-the-art analytical methods [15,16,65–74]. However, the authors viewed big data as a cultural and socio-technical phenomenon and conducted a scholarly inquiry into the social impacts of environmental big data, thereby contributing to discussions of the use of big data in society and its implications for management research. This study pays particular attention on how big data can be used to generate new sources of values and the mechanisms through which such value is fulfilled, which helps highlight an important research area, namely a new governance paradigm for sustainable development in the era of big data. Second, previous studies on the relationship between ICTs and ENGOs in China focused on the ENGOs' use of the internet and Web 2.0 technologies [1,2,4–8]. As ICTs continue to evolve to Web 3.0, which is centered around data, the recent years have seen the rise in the data and data-informed activities of ENGOs in China and other countries [20–22]. The current research falls short of capturing and analyzing this new trend. Nonetheless, as it is among the first to explore big data adoption in Chinese ENGOs, this study provides a good starting point to understand the growing role of new advanced technology in driving China's environmental activism and facilitating environmental governance. Third, this research adopts social capital theory as the theoretical foundation to analyze the social effects of big data, an emerging and new research field. Our empirical work helps to establish the usefulness and potential of social capital theory in addressing questions about how big data can reconfigure social relations and its implications for Chinese society. In doing so, it contributes to the literature on ENGOs in general by introducing social capital theory as an important theoretical lens to gain an understating of ENGOs' engagement with big data and its impacts on environmental sustainability in different contexts. Researchers could use or modify this analytical framework to examine the applications and implications of big data with regard to ENGOs in other countries. For example, is social capital created as a result of ENGOs' adoption of big data in those societies? Do the outcomes of their big data-based green social capital converge with our findings? Fourth, the existing literature has witnessed a maturation of social capital research towards a multilevel theoretical perspective concerning individuals and collectives (i.e., groups and/or organizations) [25–28,31–33,39–41,50,51].This research sheds new light on the formation of social capital at the group and organizational levels, and provides some insight into how one may identify and measure the components of social capital at the aggregate level.

While developing our research, the authors noted two limitations in their approach. First, regarding social capital, our analysis concentrated primarily on how big data helps build social capital and facilitates environmental governance. However, the authors recognize that social capital also has negative connotations [122]. The development of big data-based green social capital faces the risk of environmental injustice stemming from urban–rural inequality. Taking Green Hunan as an example, two key factors determine civic engagement in creating and sharing environmental data on a large scale. The first involves the number of ENGOs in a region. Green Hunan has partnered with numerous other local ENGOs to mobilize volunteers and promote the Riverwatcher Action Network nationwide. Chinese ENGOs are mostly urban-based and are more active in urban areas. The higher the number of ENGOs in one region, the greater the number of people likely to be motivated to become involved. The other factor concerns people's general technology skills and knowledge, such as using social media, as well as environmental consciousness and awareness. In this regard, urban citizens are at an advantage. Therefore, these two factors are in favor of urban areas, and would probably lead to an uneven playing field for public participation in environmental matters and unequal distribution of green social capital. However, this study could not explore such issues in this research, and the authors recognize that much work needs to be done to attain a balanced understanding of the impacts of big data in terms of creating social capital. Moreover, the authors largely consider the two facets of social capital separately. However, of great interest is the interrelationship between structural and cognitive capital derived from the ENGOs' big data-enabled activities. This study regards this as an important

focus for future research [45]. Finally, driven by policy support and technology integration, China's big data sector will continue to expand. ENGOs have increasingly embraced this opportunity and are equipping the civil society with the technology and skills to adopt and exploit big data. With the help of big data, Chinese ENGOs will play a more important role in creating governance practices that promote environmental protection and enable the transition to sustainability.

**Supplementary Materials:** The following are available online at http://www.mdpi.com/2071-1050/12/8/3386/s1. The file "1-Data_Figure2.zip" contains the data and codes for Figure 2; the file "2-Data_Figure4.zip" is the data source for Figure 4; the file "3-Data_Figure 5.xlsx" is used to plot Figure 5.

**Author Contributions:** Conceptualization, J.X.; methodology, J.X.; software, H.Z.; validation, H.Z.; formal analysis, J.X.; investigation, J.X.; resources, J.X. and H.Z.; writing—original draft preparation, J.X.; writing—review and editing, J.X. and H.Z. All authors have read and agreed to the published version of the manuscript.

**Funding:** This research received no external funding.

**Acknowledgments:** We would like to express sincere gratitude to Qiu Xinhua in IPE and Jiang, Xiaoyu in Green Hunan, as well as environmental activists from other ENGOs, who were generous of their time and knowledge to support our fieldwork in the summer of 2018 and follow-up interviews in the winter of 2018, 2019.

**Conflicts of Interest:** The authors declare no conflicts of interest.

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
