# Peer review of "Environmental Activism and Big Data: Building Green Social Capital in China"

_sustainability, doi:10.3390/su12083386_

Round 1

Reviewer 1 Report

Dear Authors,

the topic of social capital and environmental issues from a big data perspective is actual and interesting for a broader audience.

However, the methodological approach to the topic of examining social capital is rather unusual while using the qualitative approach, with 2 cases. additionally, added needs to e the level of analysis, because particular types of social capital can be measured on different levels of analysis.

Namely, for measuring structural and cognitive capital, there exist developed measurement scales. Authors should use them or at least extensively explain why they used a different approach. However, also using interviews, it must be added, how many interviews were done in the Method part.

However, authors should also add in the Methods, which indicators or elements of each type os social capital were measured by interviews.

Additionally, it is questionable, whether the results of two case studies can be generalized.

Wish you good luck with the article.

Reviewer 2 Report

There are some areas in which the language could be improved. Specifically:

Line 31: mutual constitutive --> mutually Line 59-60: which was mainly be attributed to the use of Line 63: you mention W3C, but do not define the abbreviation beforehand Line 78: database --> databases Line 161: IPE has dedicated to --> has been dedicated to Line 201: By virtual of --> virtue of Line 216: almost exact copy from introductory paragraph Line 334: company's responsibility, strengthen, or capacity in --> strength, or capacity in Line 349: enforcement becomes tougher in recent years Line 402 & 422: capitalize 'Figure 3' & 'Figure 4' Lines 406, 413: capitalize 'Microblogging' Line 458: integrate --> integrates Lines 470 & 486: capitalize 'Figure 5' Line 518: "cancer village" --> "cancer villages" Line 631: what the implications big --> what implications big

The authors indicate that they will only be relying on two of the three facets of social capital theory, but do not provide a clear explanation of why they choose not to include relational aspects of the theory. There are several descriptions in the article that sound as though they are referring to aspects of relational social capital, so this seems particularly confusing. A more clear explanation of the choice to not include this component, or inclusion of this component in the analysis is needed.

The authors indicate that this is a mixed methods approach case study. However, they provide no description of the approach of qualitative data analysis used (for example, did they use discourse analysis for interviews, grounded theory, etc.). A more complete description of this approach is necessary to understand how they synthesized the data streams that they described.

Figure 2 is visually challenging. In particular, it may be helpful to provide colors that offer a more clear distinction between the grey and black lines for readers.

Round 2

Reviewer 1 Report

Dear Authors,

the article is now improved and more expressive to the readers. I recommend it to be published.

Best regards

Author Response

Thank you. We very much appreciate your work in the whole review process.

This manuscript is a resubmission of an earlier submission. The following is a list of the peer review reports and author responses from that submission.

Round 1

Reviewer 1 Report

Environmental Activism and Big Data: Building Green Social Capital in China

This is an interesting and informative article that I enjoyed reading. It is by and large well written and very accessible – although in places it would benefit from some minor language editing. It is also clearly structured. Thematically, it focuses on the decentralisation of environmental policy making in China which, in addition to state government, increasingly relies on regional and local structures and seeks to engage or activate citizens.

In general terms, the article is interesting for the readership of academic journals such as Sustainability or Environmental Politics. I am not competent to comment on the article’s suitability for the Economic, Business and Management dimension of Sustainability. From my perspective as a social scientist, however, I would – despite the pieces quite well developed status – very explicitly not recommend publication of this paper. I am also sceptical about the potential of reworking this piece in order to bring it to publishable standard. The reasons for this recommendation are as follows:

Contrary to what the title of the paper suggests, this piece is neither about ‘Environmental Activism’, nor about ‘Big Data’, as commonly understood, nor about ‘Green Social Capital’. Instead it is about the efforts of the Chinese government to increase the efficiency of its environmental policy by drawing on non-state actors.

China has a highly centralised, authoritarian and oppressive government which does indeed make very large scale use of Big Data for purposes of detailed mass surveillance and control. In this article, however, none of this is even mentioned – nor is the credit system that the Chinese government has developed and is rolling out in order to enforce mass surveillance and control. In this article, the term Big Data is, rather misleadingly, being used for the collection and dissemination of information on water or air quality and environmental conditions, more generally. The collection and exploitation of personal data on citizens in contrast, is not even mentioned. And there is no form of engagement in this article with the comprehensive literature on Big Data, data mining, public surveillance, repurposing of digital trace data for agendas which citizens are neither aware of nor control etc. For an academic article this is unacceptable.

There is a comprehensive debate going on at the moment about the potentials and limitations of democratic government for the purposes of environmental policy. Many observers have recently raised doubts about the ability of democratic approaches to bring about the kind of transformation that is required in order to make modern consumer societies socially and ecologically sustainable. In this debate, China is often being referred to as an example of authoritarian approaches – perhaps – being able to deliver results which democratic systems often fail to deliver. For purposes of this article, it would be essential to embed the discussion into this debate or at least relate it to this debate. Yet, the authors are making no effort of doing so – and demonstrate no awareness of these debates, either.

The efforts of the Chinese government to make more use of decentralised structures of governance has been widely noted in the literature. Equally, there is a fairly comprehensive critical governance literature which explores, to what extent such efforts of decentralisation and ‘stakeholder engagement’ – in China and elsewhere – can really be regarded as a) the decentralisation of power and empowerment of non-state actors, and b) as conducive to the achievement of environmental goals. Again, for the purposes of this article, recognition of and engagement with these literatures would be essential. Yet, none of this is to be found in this article.

As regards the concept of social capital, this has been developed in order to investigate the condition and development of democratic systems. At a later stage it had been pointed out that non-democratic, authoritarian systems, criminal networks, terrorist associations etc. also nurture and strongly rely on social capital. This article shows no awareness of this ambiguous nature of social capital and grafts this concept rather artificially onto the subject of its enquiry. It may well be possible to apply this concept in a constructive way, but this would require much more nuanced discussion and justification.

There are other issues which one may well want to take issue with (e.g. methodological issues re interviews, data collection, data analysis). But in view of these major academic deficits, this paper is – from a social science perspective – not suitable for publication. Its misuse of key concepts such as environmental activism, big data and green social capital is – perhaps unintentionally – strongly ideological in that it gives Chinese policy a gloss of democratic participation and legitimacy which is highly problematic. Indeed major sections of this paper read a bit like the Chinese government presenting, celebrating – and hugely simplifying – its policy approaches: identification, reporting and responding to black, filthy and smelly water bodies.

Things are quite more complex than this; but nevertheless, I much enjoyed reading this informative and well written paper.

Reviewer 2 Report

The paper presents an interesting case study to explore how the adoption of big data by Chinese environmental NGOs contributes to the creation of social capital by fostering cooperation between different actors. The paper focuses on two specific initiatives by the IPE ENGO in which big data was used to promote environmental governance. After explaining the importance of ICTs and big data for the operation of ENGOs, the paper turns to explaining the concept of social capital and the methodology used. Then, it explores how each of the chosen initiatives contribute to the creation of structural and cognitive social capital.

In general terms, the paper is presented in a clear and compelling way, provides an interesting and informative case and problem, and I believe it can be understood by a wide range of audiences. Nevertheless, I have some specific comments:

The nature and rationale behind the methods for data analysis are not presented in enough detail to allow the reader to understand what they are, why they were chosen, and what findings they will uncover (lines 200-202). This is particularly the case with the “quantitative analysis using Python package Network X” (lines 200-202 and 277-279), which might not be familiar for the wide range of readers of Sustainability. In section 4.2.2., the paper could benefit a lot from discussing the literature that explores empirically consumer’s understandings of sustainability and how it influences their buying decisions (i.e. https://onlinelibrary.wiley.com/doi/full/10.1111/j.1470-6431.2011.01045.x or https://www.emerald.com/insight/content/doi/10.1108/13612021211265863/full/html) For the structural ties in the Blue Map case (section 4.3.1), I wonder why wasn’t a standardized measure (such as reports per inhabitants) used? Taking the absolute number of reports might be misleading because of the differences of population between provinces. The nature of the cognitive capital created through the Blue Map initiative (section 4.3.2) is not very clear. The section describes several contextual and specific elements, but it could be much clearer in explaining the “shared representations, interpretations, and systems of meaning among actors that play a motivational role in fostering and facilitating collective actions of people or groups within a network” (lines 144-146) beyond (or around) the shared objective of identifying the black, filthy and smelly water bodies. I would suggest to the authors to describe more clearly/in more detail the similarities and differences between the two initiatives they present. This could provide interesting insights into how IPE creates social capital networks depending on the actors and goals involved, for example. In general, I would also suggest to the authors to evaluate whether including more evidence from their data (i.e. interview or document quotes, quantitative figures, reports of activities, and so on), could give more strength to their interpretations on the creation of social capital.

Reviewer 3 Report

Dear authors,

The proposed article is interesting for readers and as such well prepared.

However, there is one thing that should be improved. Missing is the part, entitled Discussion. It provides an added value for readers.

Othervise is the article suitable for publication.

Good luck with the article!